# Impact of Synbiotic Intake on Liver Metabolism in Metabolically Healthy Participants and Its Potential Preventive Effect on Metabolic-Dysfunction-Associated Fatty Liver Disease (MAFLD): A Randomized, Placebo-Controlled, Double-Blinded Clinical Trial

**DOI:** 10.3390/nu16091300

**Published:** 2024-04-26

**Authors:** Aakash Mantri, Anika Köhlmoos, Daniela Stephanie Schelski, Waldemar Seel, Birgit Stoffel-Wagner, Peter Krawitz, Peter Stehle, Jens Juul Holst, Bernd Weber, Leonie Koban, Hilke Plassmann, Marie-Christine Simon

**Affiliations:** 1Institute of Nutrition and Food Science, Nutrition and Microbiota, University of Bonn, 53115 Bonn, Germany; aamantri@uni-bonn.de (A.M.); akoehlmoos@zoologie.uni-kiel.de (A.K.); wseel@uni-bonn.de (W.S.); 2Institute for Genomic Statistics and Bioinformatics, University Hospital Bonn, 53127 Bonn, Germany; pkrawitz@uni-bonn.de; 3Center for Economics and Neuroscience, University of Bonn, 53113 Bonn, Germany; daniela.schelski@uni-bonn.de (D.S.S.); bernd.weber@ukbonn.de (B.W.); 4Institute of Experimental Epileptology and Cognition Research, University of Bonn, 53113 Bonn, Germany; 5Institute of Clinical Chemistry and Clinical Pharmacology, University Hospital Bonn, 53127 Bonn, Germany; birgit.stoffel-wagner@ukbonn.de; 6Institute of Nutrition and Food Science, Nutritional Physiology, University of Bonn, 53115 Bonn, Germany; pstehle@uni-bonn.de; 7Novo Nordisk Foundation Center for Basic Metabolic Research, Department for Biomedical Sciences, University of Copenhagen, DK-2200 Copenhagen, Denmark; jjholst@sund.ku.dk; 8Institut Européen d’Administration des Affaires (INSEAD), 77300 Fontainebleau, France; leonie.koban@cnrs.fr (L.K.); hilke.plassmann@insead.edu (H.P.); 9Control-Interoception-Attention Team, Paris Brain Institute (ICM), 75013 Paris, France; 10Lyon Neuroscience Research Center (CRNL), Centre National de la Recherche Scientifique (CNRS), Institut National de la Santé et de la Recherche Médicale (INSERM), Université Claude Bernard Lyon 1, 69500 Bron, France

**Keywords:** synbiotics, prevention, ALT, microbiome, gut–liver axis, MAFLD, randomized controlled trial, metabolic healthy participant

## Abstract

Synbiotics modulate the gut microbiome and contribute to the prevention of liver diseases such as metabolic-dysfunction-associated fatty liver disease (MAFLD). This study aimed to evaluate the effect of a randomized, placebo-controlled, double-blinded seven-week intervention trial on the liver metabolism in 117 metabolically healthy male participants. Anthropometric data, blood parameters, and stool samples were analyzed using linear mixed models. After seven weeks of intervention, there was a significant reduction in alanine aminotransferase (ALT) in the synbiotic group compared to the placebo group (−14.92%, CI: −26.60–−3.23%, *p* = 0.013). A stratified analysis according to body fat percentage revealed a significant decrease in ALT (−20.70%, CI: −40.88–−0.53%, *p* = 0.045) in participants with an elevated body fat percentage. Further, a significant change in microbiome composition (1.16, CI: 0.06–2.25, *p* = 0.039) in this group was found, while the microbial composition remained stable upon intervention in the group with physiological body fat. The 7-week synbiotic intervention reduced ALT levels, especially in participants with an elevated body fat percentage, possibly due to modulation of the gut microbiome. Synbiotic intake may be helpful in delaying the progression of MAFLD and could be used in addition to the recommended lifestyle modification therapy.

## 1. Introduction

Because the gut is directly connected to the liver via the portal vein, known as the gut–liver axis, the gut microbiome and its modulation in the prevention of MAFLD are of particular interest [1,2]. MAFLD, characterized by excessive accumulation of fat in the hepatic tissue, is the most common liver disease in Western populations. Its prevalence ranges from 17–46% depending on sex, age, ethnicity, and method of diagnosis [3], and is steadily increasing—as are its underlying risk factors, especially visceral obesity, and type 2 diabetes mellitus (DMT2) [4]. MAFLD includes a number of diseases, such as simple steatosis, also termed non-alcoholic fatty liver disease (NAFLD), non-alcoholic fatty liver (NAFL), and non-alcoholic steatohepatitis (NASH), which can progress to fibrosis, cirrhosis, and hepatocellular carcinoma (HCC) [5].

Several recent studies have demonstrated correlations between bacterial composition and distinct taxa and MAFLD or NASH and explored the potential mechanisms by which the gut microbiota may regulate MAFLD and NASH [6,7,8], as it has been shown in the development of obesity [9] and DMT2 [10,11]. The suggested mechanisms are as follows: the microbiome might regulate MAFLD and NASH by contributing to obesity and microbial dysbiosis as underlying risk factors [9,12,13], and microbial dysbiosis leads to increased gut permeability [14], allowing dysbiotic bacteria and their metabolites to translocate to the liver through a disrupted gut barrier [15,16]. Therefore, it is essential to further characterize the role of the microbiome–gut–liver axis in the development and prevention of MAFLD [17]. 

The microbiome can be modulated by synbiotics [18,19], prebiotics, and probiotics [20], or at present, even postbiotics [21]. Synbiotics are known to have positive effects on the host’s health, with synergistically greater effects than pro- or prebiotics alone [22]. Furthermore, previous studies have shown that synbiotics may also have therapeutic potential for obesity [23,24,25], insulin resistance [26], DMT2 [27], inflammatory bowel syndrome [5,28,29], and MAFLD [23,24,25,30,31,32]. Additionally, synbiotics have been used as adjuncts before and after bariatric surgery [6]. Despite their microbiome-modulating properties [33,34], it remains unknown whether synbiotics can be used to prevent obesity and MAFLD.

Although the use of synbiotics as supportive therapy for a variety of non-communicable diseases is being increasingly investigated, only a few studies to date have examined the preventive effects of synbiotics. Therefore, the aim of our study was to investigate the preventive potential of synbiotics in metabolically healthy participants and to examine their possible association with microbiota composition in a 7-week human intervention study. Because of the direct link between the gut and the liver, we focused on the effects of liver enzymes as biomarkers of MAFLD. 

## 2. Material and Methods

### 2.1. Ethical and Open Science Consideration

Written informed consent was obtained from all participants at the beginning of the study. The study protocol was approved by the Ethics Committee of the University Clinic Bonn (number 347/18, Approval Date: 17 December 2018) and was conducted in accordance with the guidelines of the 1964 Declaration of Helsinki and its later amendments. The trial was pre-registered at Open Science Framework (https://osf.io/utsn4 (accessed on 5 March 2024)), with a detailed description of the overall protocol.

### 2.2. Study Design and Intervention

A 1:1 randomized, placebo-controlled, double-blinded longitudinal study was conducted between March and November 2019 at the University Hospital Bonn, Germany. Participants and investigators were not aware of the allocated group, outcome assessors. The participants attended two sessions in which anthropometric data, blood, and fecal samples were collected. Blood samples were analyzed for parameters of lipid and glucose metabolism as well as liver enzymes, inflammation markers, serotonin, and amino acids. The fecal samples were used to analyze the microbiome.

After the first session, the participants were randomly assigned to two groups (synbiotic (SYN) vs. placebo (PLA)). The synbiotic group received a dietary synbiotic supplement containing 2 × 10^9^ cfu probiotic bacteria of five strains (*Bifidobacterium lactis*, *Lactobacillus acidophilus*, *Lactobacillus casei*, *Lactobacillus salivarius*, and *Lactococcus lactis*) and inulin from agave as a prebiotic. The placebo group received microcrystalline cellulose (MCC) that was identical in packaging, form, and taste. The participants were instructed to consume 2 g of their supplement dissolved in water each day at the same time for seven weeks without otherwise changing their dietary habits and physical activity.

### 2.3. Participants

The study population consisted of 117 male participants (20–60 years old, with a body mass index [BMI] of 20–34 kg/m^2^) (see Appendix A for details), according to the inclusion criteria as listed in the pre-registration (https://osf.io/utsn4 (accessed on 5 March 2024)). Only participants without dietary restrictions (such as being vegetarian or vegan), without food allergies and intolerances, and who did not take any hormonal medication or antibiotics (as these have an independent impact on the human gut microbiota) were included in this study. 

Of the initial 117 participants, 31 were excluded based on pre-registered exclusion criteria. Reasons for exclusion were antibiotic treatment at any time during the intervention (*n* = 8), changes in medical conditions (e.g., gastroenteritis) or treatment that potentially impacted the gut microbiota or blood parameters (*n* = 6), reporting a change in dietary habits (*n* = 1), and not attending the post-intervention session (*n* = 1). In contrast to the pre-registration, we changed the minimum compliance criterion for supplement intake during the intervention from 50% to 95%, which is more conservative and in line with the standards in nutrition science. Thus, an additional 15 participants were excluded because they took the supplement less than 95% of the intervention time (Appendix A). All criteria were assessed using self-reporting. Analysis was based on a sample of 86 participants. The synbiotic group included 45 participants, whereas the placebo group included 41 participants. 

For the analyses stratified by body fat percentage, additional participants (*n* = 5) were excluded due to technical issues with body fat assessment, leaving 81 participants. Table 1 shows the baseline characteristics of the study population. Follow-up measurements were planned at 6, 12, 18, and 24 months. However, due to the COVID-19 pandemic, it was possible to carry out follow-up measurements only in one in-person visit, which caused a delay of assessment by half a year. In total, 45 participants completed the follow-up measurements (Appendix A). 

### 2.4. Anthropometrics and Categorization

Weight and height were measured, and BMI was calculated using the standard formula: BMI = weight [kg]/height [m]^2^. Body weight and body fat proportion were determined using a medical-grade bioimpedance scale (Tanita Europe BV, Amsterdam, The Netherlands). The participants were classified as having a physiological or elevated body fat percentage depending on age and body fat percentage, based on Gallagher et al. [35], with the following criteria for stratification:
Age 20–39: physiological body fat mass: ≤20%, elevated body fat mass: >20%.Age 40–59: physiological body fat mass: ≤22%, elevated body fat mass: >22%.Age 60–79: physiological body fat mass: ≤25%, elevated body fat mass: >25%.

### 2.5. Dietary Intake

Dietary intake was recorded by the participants who were instructed to fill in a food journal for three consecutive days before their first and second sessions. Data were transferred to the nutritional software EBISpro 2016.

### 2.6. Blood Sample Processing and Analysis

Blood samples were collected after 12 h of over-night fast. The samples were centrifuged within 30 min of blood collection, and plasma/serum aliquots were stored at −80 °C until further use. All biochemical analyses were performed according to manufacturer’s instructions. Briefly, blood parameters were analyzed using the Roche/Hitachi Cobas c system (Roche Diagnostics, Mannheim, Germany). 

Triglyceride (TG) and total cholesterol (TC) levels were measured using enzymatic colorimetric assays. Low-density lipoprotein (LDL) cholesterol and high-density lipoprotein (HDL) cholesterol levels were determined by homogenous enzymatic colorimetry. Aspartate aminotransferase (AST) and ALT were measured using a photometric assay, gamma-glutamyl transferase (GGT) was measured using a homogenous enzymatic assay, and alkaline phosphatase (PAL) was measured using a colorimetric assay. High-sensitivity C-reactive protein (hs-CRP) levels were determined using a particle-enhanced turbidimetric immunoassay. Glucose was measured using the hexokinase method, and insulin and interleukin-6 (IL-6) levels were measured using an ElectroChemiLuminescence ImmunoAssay. HbA1c levels were determined using ion-exchange high-performance liquid chromatography (Bio-Rad, Hercules, CA, USA). The homeostasis model assessment index for insulin resistance (HOMA-IR) was calculated using the following formula: HOMA-IR = Fasting insulin (mU/L) × fasting glucose (mg/dL)/405. Serotonin levels were measured by HPLC (Chromsystems Instruments & Chemicals GmbH, Gräfelfing, Germany). Glucagon-like peptide-1 (GLP-1) levels were measured using radioimmunoassay. Amino acids were analyzed using high-performance liquid chromatography, according to standard operating procedures.

### 2.7. Gut Microbiome Sample Processing and Analysis

Fecal samples were collected within 24 h before each session according to a standard operation procedure and immediately stored at −80 °C until further analysis. DNA was prepared from stool samples using the QIAamp PowerFecal DNA Kit according to the manufacturer’s instructions (Qiagen, Hilden, Germany). In brief, mechanical lysis of stool samples was performed using Bead Tubes with a 0.7 mm Dry Garnet. High-throughput 16S amplicon sequencing was performed on 233 samples from the 16S V3V4 region using the primer combination 341f-806bR. 

### 2.8. Statistical Analysis

All statistical analyses were performed using R Studio (version 3.6.2, Boston, MA, USA). For all analyses, a *p* value < 0.05 was considered statistically significant. Continuous data were expressed as mean ± standard deviation (SD), and categorical variables were expressed as frequencies. The distribution of continuous variables was tested using the Shapiro–Wilk test. To test for differences between the treatment groups at their first appointment, an unpaired Student’s *t*-test was conducted for continuous variables, the Mann–Whitney test for non-parametric variables, and Pearson’s chi square test for categorical variables. To rule out the possibility that changes in blood parameters were due to differences in dietary intake, the *t*-test or Wilcoxon test was used to compare the relative change in the intake of energy, carbohydrates, protein, and fat, as well as body weight, BMI, and fat mass between the groups. To confirm the intervention effect on ALT, a linear model (effect of group) was used on the relative change in ALT. Relative change is defined as ((Session 2 − Session 1)/Session 1) × 100. Age and fat mass have been shown to affect ALT concentrations significantly and have been included as covariates.

An additional analysis was performed to check the absolute change in ALT concentration using linear mixed models (effect of group, visit, and group × visit interaction at each time-point in the time-course analysis). If the model assumptions were not met, data was log-transformed. Age and fat mass were included as covariates. The first session and placebo groups were selected as references. Moreover, in all the regression analyses, the residuals were checked for deviations from the normal distribution and homoscedasticity. A post hoc analysis on the ALT parameter showed a power of 100%. 

For the gut microbial compositional analysis and to confirm the intervention effect on the gut microbiome, a linear mixed model was used to test the differences between the groups regarding microbial α-diversity (Shannon index and Faith phylogenetic diversity). Analyses of β-diversity (Bray–Curtis Distance) additionally included the baseline α-diversity and the two-way interactions (group × α-diversity). The placebo group served as the reference group. 

The 16S rRNA sequencing data preprocessing was carried out as follows: QIIME2 (Quantitative Insights into Microbial Ecology; version 2023.5) [36] was used for all preprocessing steps. The resulting 300 bp paired-end reads from the MiSeq analysis were assembled using DADA2 [37]. DADA2 was also used for quality filtering of paired-end reads, which is based on a quality score of >30 and the removal of mismatching barcodes. ASVs (amplicon sequence variants) generated from DADA2 were used for further analysis. A phylogenetic tree was created using these ASVs. Finally, the silva taxonomy database [38,39] was used for the taxonomic assignment of sequences at all taxonomy levels and their relative abundances were estimated for each level. For the diversity analysis, rarefaction at a sampling depth of 42,251 sequences was performed. Thus, 24 samples were dropped for the diversity analysis. The QIIME2 package was used to calculate alpha diversity metrics such as the Shannon index and Faith and beta diversity metrics, such as Jaccard distance.

A longitudinal taxonomy analysis was performed on the gut microbiome of 86 participants. For the taxonomy analysis, we considered 121 adequately abundant taxa using the criterion that each taxon is present in at least 20% of the samples [40]. For the regression analysis of the gut bacterial taxa, a negative binomial and zero-inflated mixed model [41] was used. This model addresses zero-inflation issues in some microbiome taxa. The model consists of two steps: the first part is a logistic model for predicting excess zeros, and the second is a negative binomial distribution for over-dispersed counts. We used this model (effect of group, session, and group × session) to identify the microbial taxa that were significantly changed by the intervention. Age and fat mass were used as the covariates. The model was adjusted for the different numbers of sequences in each sample. The model includes random effects that correlate with repeated sampling of microbiota in the same individual. The model was applied individually to each taxon. To correct for multiple testing, a false discovery rate (FDR)-adjusted *p* < 0.05 was selected for the associated genus. 

## 3. Results

A total of 86 male participants with an average age of 32 ± 11 years were included in the analysis (Table 1). There were no significant differences at baseline between the synbiotic and placebo groups in terms of anthropometric data, liver enzymes such as ALT, and other metabolic parameters of glucose or lipid metabolism. Furthermore, there were no significant differences in dietary intake between the two groups at baseline or after the intervention (Table 2). Since participants were instructed to adhere to an isocaloric diet, after the seven-week intervention period, there were no differences in the relative change in weight, BMI, or fat mass. Likewise, no significant differences in the relative change in energy intake, carbohydrate intake, protein intake, or fat intake were observed (Table 2).

Forty-five participants completed a one-year follow-up. No significant differences were observed between the synbiotic and placebo groups at the follow-up appointment. Since many participants reported changes in their dietary and/or exercise behaviors due to the COVID-19 pandemic during follow-up, the data are not representative and were not used for further analyses (Appendix A).

After the 7-week intervention, the ALT concentration was significantly lower in the synbiotic group than in the placebo group (Figure 1A), while the body weight remained stable (Table 2). This effect was driven by the participants in the elevated body fat subgroup, as their ALT concentration was significantly lower than that in the placebo group (Figure 1B), while the ALT concentration in participants with physiological body fat remained stable. An additional analysis on ALT concentration using absolute change showed similar results. The ALT concentration was significantly lower in the synbiotic group than in the placebo group as well. This effect was also driven by the participants in the elevated body fat subgroup, while the ALT concentration in participants in the physiological body fat subgroup remained stable.

As expected, the gut microbiome composition at baseline showed individual variation (Shannon index: 0.57 min., 0.89 max.) but was similar between intervention groups (Jaccard distance: 0.9 ± 0.1). Upon the intervention, the gut microbiome composition remained stable and was not affected by age, fat mass, or baseline microbiome diversity (Figure 2A).

The taxonomic analysis at the genus level revealed changes in some genera within each group upon intervention (synbiotic: 22 genera, placebo: 27 genera), whereas the most abundant genus, the core microbiome, remained stable (Figure 2B). Furthermore, the intervention led to a significant increase in nine genera belonging to the phyla Firmicutes (*Merdibacter*, *Lactobacillus*, *Lactococcus*, the *Eubacterium eligens* group, the *Eubacterium ruminantum* group, and *Veillonella*), Actinobacteria (*Adlercreutzia*), and Proteobacteria (*Oxalobacter*) in the symbiotic group compared to the placebo group (Figure 2C). Additionally, a significant decrease in the abundance of ten genera belonging to the phyla Firmicutes (*Faecalitalea*, *Agathobacter*, the *Ruminococcus gauvreauii* group, *UCG-009*, *Oscillospira*, *Subdoligranulum*, *Phocea*, *Candidatus Soleaferrea*, and the *Eubacterium brachy* group) and Actinobacteria (*Olsenella*) were identified. 

Although the microbiome composition of participants with elevated body fat was altered, that of participants with physiological body fat was unaffected by the intervention (Figure 3A). Moreover, the alterations in the elevated body fat group due to the intervention depended on baseline microbial diversity (Figure 3B, Appendix A). 

The taxonomic analysis at the genus level revealed that three genera that were significantly increased across all participants were also significantly increased in the elevated body fat group, and two of these genera were significantly decreased in the elevated body fat group (Figure 3C–F).

Additionally, changes in ALT were positively associated with the change in abundance of bacterial genera such as the *Ruminantium* group (family: Eubacterium), the *Clostridia vadin BB60* group, *UCG−009*, and *Negativibacillus* and negatively associated with the change in abundance of *Lactobacillus*, the *Methylpentosum* group, *Merdibacter*, *Veillonella,* the *Ruminococcos gnavus* group, *Faecalitalea*, and *Akkermansia*.

## 4. Discussion

In this randomized, placebo-controlled, double-blind intervention study, a seven-week intake of a specific synbiotic supplement resulted in a reduction in ALT concentration in metabolically healthy participants, highlighting the possible impact of synbiotics in the disease prevention of obesity and MAFLD due to their potential microbiome-modulating properties. This finding was more prominent in participants with higher body fat percentages. Thus, participants at a risk of developing MAFLD and metabolic syndrome may benefit from synbiotic interventions.

No previous study has investigated the preventive effect of synbiotic administration on liver enzymes in metabolically healthy participants, in general or specifically on ALT, and its prospective benefit in the prevention of MAFLD. Previous studies have focused on patients with MAFLD or, in particular, NASH [17]. In these studies, the administration of synbiotics resulted in delayed disease progression [42], showing a reduction in ALT [32] similar to that observed in the present trial, and there was a significant decrease in AST [32]. Thus, the present study shows that the intake of synbiotics may already reduce ALT levels in metabolically healthy individuals, and thus represents a possibility to delay the onset of disease in addition to delaying disease progression. These beneficial effects were achieved without causing any side effects to the participants. Furthermore, this indicates that an intervention with synbiotics might be used as a supportive supplement to the recommended lifestyle modification therapies, such as diet or increased physical activity with the goal of weight loss [3,43]. In this study, a notable reduction in ALT levels was observed even in the absence of detectable changes in diet, body weight, and composition. This suggests that the observed alterations in ALT levels can be attributed exclusively to the effects of the administered synbiotic intervention. Therefore, it might be worthwhile to investigate this effect further, especially given that no pharmacological agent has been approved so far for the treatment of MAFLD [44], owing to insufficient improvement in liver histology or the occurrence of adverse effects, neither of which has been reported for a synbiotic intervention. Only recently, a study [45] found that resmetirom improved fibrosis in 966 patients with NASH and fibrosis of stage F1B, F2, or F3. 

The change in microbiome composition and the increase in probiotic strains of *Lactobacillus* and *Lactococcus* during the intervention showed that the intervention modified the gut microbiome and thus likely drives the effects on liver metabolism via the microbiome–gut–liver axis. An increase in *Lactobacillus* and its negative correlation with ALT levels indicates that *Lactobacillus* plays an important role in attenuating the progression of MAFLD, as reported previously [46,47]. 

Furthermore, baseline microbial diversity seemed to affect the success of the intervention, resulting in improved metabolism. Thus, as previously mentioned, gut bacteria such as *Veillonella* and *Merdibacter* significantly increased after the intervention in the synbiotic group and were associated with a decrease in ALT levels. In addition, *Akkermansia* was found to be negatively correlated with ALT and has been proposed as a candidate probiotic for the treatment of various diseases [48]. Additionally, the intake of synbiotic bacteria, such as *Akkermansia muciniphila*, was used to improve the conditions of MAFLD participants [49]. This trend has also been observed in various clinical studies that have evaluated gut microbial alterations after bariatric surgery [50].

The study has several strengths. First, this nutritional intervention study was conducted according to the gold standard for a randomized, placebo-controlled, double-blind study (RCT). Second, all blood parameters were measured, fecal microbiome analysis was conducted using standardized methods, and measurements were part of the laboratory routine with strict quality controls. Furthermore, for an RCT, a rather high number of 86 participants were included in the analyses. Lastly, the synbiotic used fulfilled all safety criteria, rendering the risk of adverse effects low from the beginning of the study, which was subsequently confirmed because no participants reported any side or adverse effects. 

One limitation of this study was that the population consisted of only male participants. Therefore, differences in the metabolism of men and women due to hormonal influences, especially gonadal hormones and the sex chromosome complement, could be avoided. Male hormone levels are rather stable, and results from studies with male subjects are potentially more reliable for comparison [50,51]. An important direction for future research would be to test the effects of synbiotics on the gut–liver–axis in female participants, taking the hormonal effects into account. 

Another limitation might be that the participants were metabolically healthy, meaning that long-term studies are necessary to strengthen conclusions on the preventive effects. However, the results from participants with MAFLD showed improvement after the consumption of probiotics [20,52]. Therefore, our results in metabolically healthy individuals complement those in diseased individuals. Furthermore, we planned and conducted a follow-up study after 6, 12, 18, and 24 months to determine long-term changes. Unfortunately, owing to the COVID-19 pandemic, participants may have reported lifestyle changes, especially regarding dietary behavior and physical activity [53,54,55].

## 5. Conclusions

In conclusion, the intake of a specific synbiotic supplement led to a reduction in ALT levels, especially in participants with an elevated body fat percentage, and did not result in any adverse effects. This reduction in ALT was possibly due to the potential microbiome-modulating properties of synbiotic supplements. Since aminotransferase levels are often elevated in MAFLD, and ALT is the most specific liver enzyme, the findings indicate that apart from the properties shown to delay the onset, the intake of synbiotics might be an effective supplement in MAFLD, in combination with the recommended lifestyle modifications, helping to delay the progression of the disease. However, further studies are needed to investigate the impact of synbiotics as well as pro- and prebiotics on liver metabolism, the microbiome–gut–liver axis, and their potential in the prevention and treatment of liver diseases.

## Figures and Tables

**Figure 1 nutrients-16-01300-f001:**
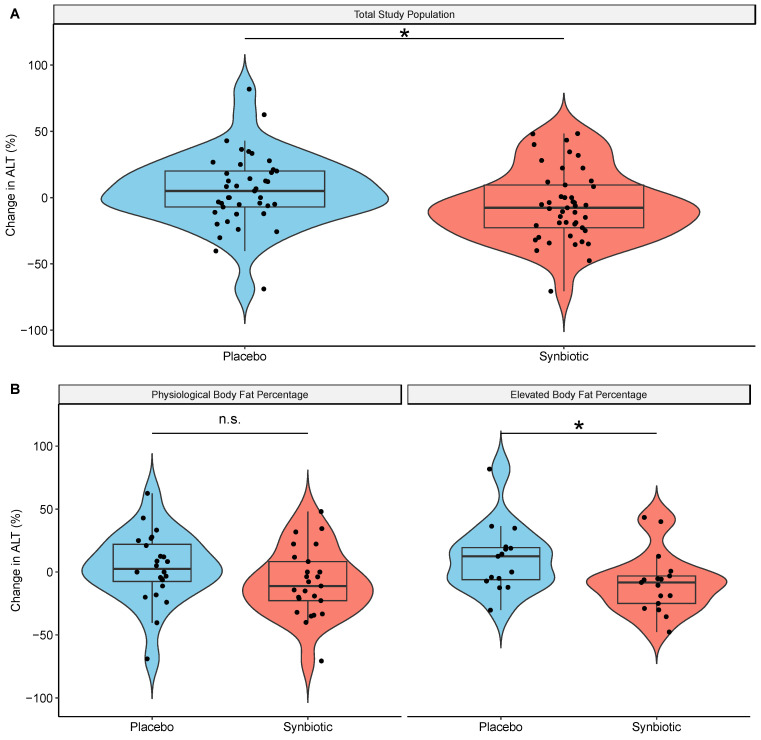
Intervention effects on ALT. (**A**) Box-plot showing relative change in ALT (%) in each intervention group. (**B**) Intervention effects on ALT stratified by body fat. Box-plot showing relative change in ALT (%) in each intervention group. Significance of effect on ALT is determined by linear mixed model. *: significant (*p* < 0.05); n.s.: non-significant (*p* > 0.05).

**Figure 2 nutrients-16-01300-f002:**
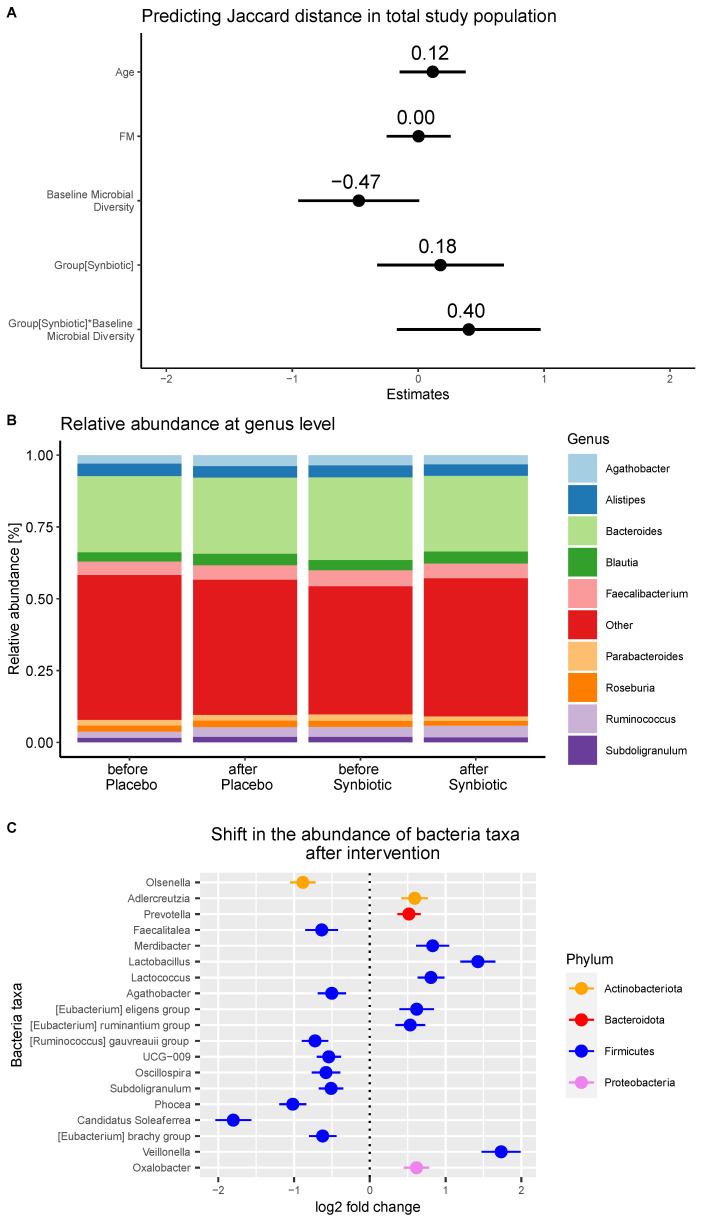
(**A**) Estimate plots from analyses of change in gut microbiome composition. Data analyzed using linear mixed model. (**B**) Relative taxonomy abundance at genus level (prevalence: 0.60, detection threshold: 0.01) in each group/time-point. (**C**) Significant changes in taxonomy abundances.

**Figure 3 nutrients-16-01300-f003:**
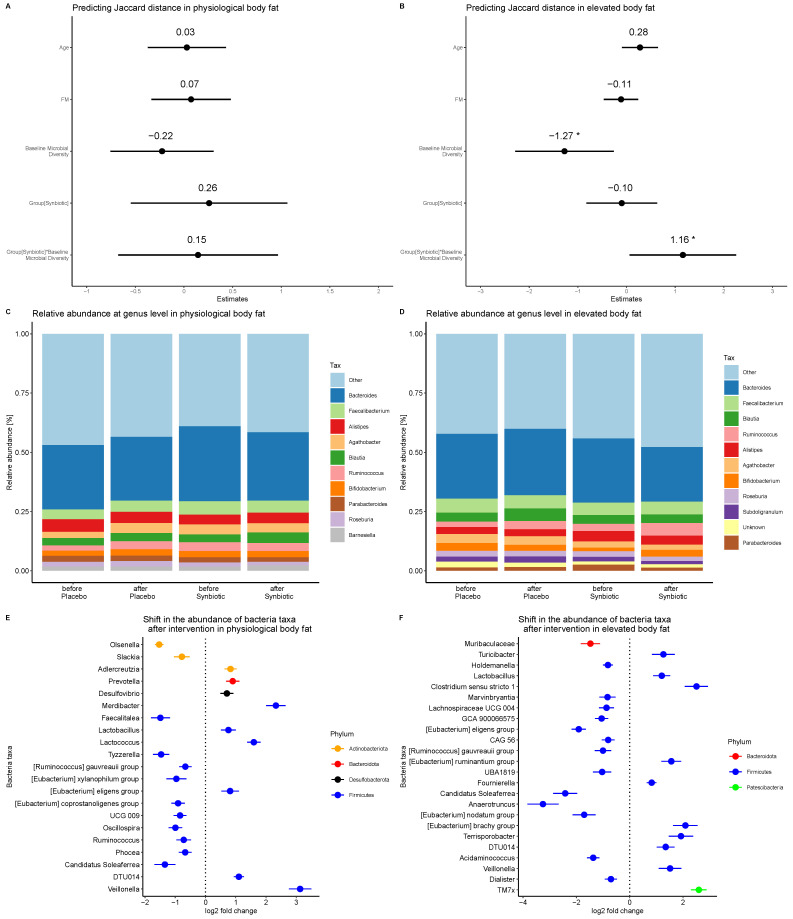
Estimate plots from analyses of changes in gut microbiome composition in (**A**) physiological body fat percentage group and (**B**) elevated body fat percentage group. Data analyzed using linear mixed model. Relative taxonomy abundance at genus level (prevalence: 0.60, detection threshold: 0.01) in each group/time-point in (**C**) physiology and (**D**) elevated body fat group. Significant changes in the taxonomy abundance in (**E**) physiology and (**F**) elevated body fat group. *: significant (*p* < 0.05).

**Table 1 nutrients-16-01300-t001:** Baseline characteristics of the study population.

	Total *n* = 86	SYN *n* = 45 (52.3%)	PLA *n* = 41 (47.7%)	*p* Value
Age (years)	32 (20, 60)	33 (20, 58)	32 (20.4, 60.1)	0.953
Height (cm)	181.38 (166.5, 198.4)	180.44 (166.5, 192.0)	182.41 (167.1, 198.4)	0.162
Weight (kg)	84.67 (64.3, 124.3)	84.18 (68.7, 124.3)	85.22 (64.3, 123.1)	0.607
BMI (kg/m^2^)	25.74 (20.5, 33.7)	25.84 (20.6, 33.7)	25.62 (20.5, 33.6)	0.747
Fat mass (%)	19.49 (11.6, 33.2)	19.41 (11.6, 33.2)	19.57 (12.5, 32.1)	0.888
Physiological body fat	49 (57.0%)	25 (51.0%)	24 (49.0%)	0.853
Elevated body fat	32 (37.2%)	17 (53.1%)	15 (46.9%)	
TG (mg/dL)	99.57 (36.0, 292.0)	95.04 (36.0, 292.0)	104.54 (40.0, 283.0)	0.149
Total chol (mg/dL)	172.51 (107.0, 300.0)	166.80 (107.0, 300.0)	178.78 (108.0, 300.0)	0.051
LDL chol (mg/dL)	108.06 (47.0, 237.0)	103.69 (47.0, 237.0)	112.85 (56.0, 237.0)	0.119
HDL chol (mg/dL)	49.51 (28.0, 85.0)	49.69 (28.0, 74.0)	49.32 (28.0, 85.0)	0.872
AST (U/L)	24.59 (12.0, 72.0)	25.49 (12.0, 72.0)	23.61 (13.0, 47.0)	0.983
ALT (U/L)	29.76 (12.0, 72.0)	31.07 (13.0, 144.0)	28.33 (12.0, 119.0)	0.411
GGT (U/L)	21.21 (8.0, 72.0)	22.71 (8.0, 72.0)	19.56 (10.0, 54.0)	0.232
PAL (U/L)	64.13 (35.0, 110.0)	65.49 (37.0, 110.0)	62.63 (35.0, 95.0)	0.369
Glucose (mg/dL)	91.67 (75.0, 136.0)	92.00 (76.0, 136.0)	91.32 (75.0, 105.0)	0.832
HbA1c (%)	5.05 (4.3, 6.7)	5.08 (4.3, 6.7)	5.02 (4.6, 5.5)	0.856
Insulin (mU/L)	8.98 (2.8, 19.9)	8.80 (2.9, 19.9)	9.19 (3.1, 18.3)	0.557
HOMA-IR	2.08 (0.6, 5.7)	2.05 (0.5, 5.7)	2.11 (0.6, 4.2)	0.565
GLP-1 (pM)	17.14 (7.0, 48.0)	17.42 (7.0, 48.0)	16.83 (8.0, 28.0)	0.979
hs-CRP (mg/L)	0.90 (0.3, 5.1)	0.81 (0.3, 3.7)	1.00 (0.3, 5.1)	0.258
IL-6 (pg/mL)	1.90 (1.5, 4.4)	1.93 (1.5, 3.8)	1.87 (1.5, 4.4)	0.882

Continuous data are expressed as mean (min, max) and categorical variables as frequencies (%).

**Table 2 nutrients-16-01300-t002:** Mean relative change ± SD of anthropometric measures and macronutrient intake from session 1 to session 2.

	SYN *n* = 45	PLA *n* = 41	*p* Value
Weight (%)	−0.02 ± 1.92	0.03 ± 1.84	0.894
BMI (%)	−0.02 ± 1.92	0.03 ± 1.84	0.894
Fat mass (%)	−2.34 ± 8.15	−1.51 ± 8.54	0.597
Energy intake (%)	−5.50 ± 27.16	−6.54 ± 20.98	0.845
Carbohydrate intake (%)	−1.43 ± 33.45	−8.55 ± 22.42	0.250
Protein intake (%)	−4.45 ± 33.23	−7.38 ± 26.43	0.979
Fat intake (%)	−8.26 ± 32.08	−3.81 ± 28.68	0.413

## Data Availability

The study was pre-registered at Open Science Framework (OSF; https://osf.io/utsn4 (accessed on 5 March 2024)). The data described in the manuscript, codebook, and analytical code may be shared only upon official request because of data protection regulations.

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
