# Peer review of "Impact of Synbiotic Intake on Liver Metabolism in Metabolically Healthy Participants and Its Potential Preventive Effect on Metabolic-Dysfunction-Associated Fatty Liver Disease (MAFLD): A Randomized, Placebo-Controlled, Double-Blinded Clinical Trial"

_nutrients, 2024, doi:10.3390/nu16091300_

Round 1

Reviewer 1 Report

Comments and Suggestions for Authors

I believe that the authors' study was carried out correctyl, considering all the obstacles encountered due to the Covid 19 pandemic. I consider the study to be highly original and to lay the foundations for future projects aimed to explore the use of synbiotics in the prevention and treatment of MAFLD. Finally, I consider a strong point of the work to have underlined the limitations of the study. For this reason, I think that the work can be accepted in its current form, after few correction in the text, listed here:

Line 68: missing space

Line 71-72: remain unknown is written twice

Line 97: There are two round brackets

Line 126: six is written in letters, change to number

Line 139-149: not aligned

Line 253: there is one parenthesis too many

Reviewer 2 Report

Comments and Suggestions for Authors

In this article the authors investigated the effects of a seven-week intake of specific symbiotic supplements on liver health and gut microbiome.

The text is well written, although there are minor misspellings and errors in the text. The use of English language is appropriate.

There are a few suggestions for the improvement of the text listed below:

Line 24, I would suggest to refrain from using references in the abstract

Line 31 and further Please use ALT for alanine aminotransferase and AST for aspartate aminotransferase

Line 62 “fore, it essential to further characterize” to “fore, it is essential to further characterize”, and other misspellings through the text

Line 142 is this a new paragraph or subheading?

Line 188 table title is not clear enough, is it after the 7 weeks? Also, other table and figure legend need improvement, they need to be understood without reading the text

Line 266 it would be prudent to mention the newly approved therapy for NASH

Comments on the Quality of English Language

listed above
